# Pencil Beam Scanning Bragg Peak FLASH Technique for Ultra-High Dose Rate Intensity-Modulated Proton Therapy in Early-Stage Breast Cancer Treatment

**DOI:** 10.3390/cancers15184560

**Published:** 2023-09-14

**Authors:** Grant Lattery, Tyler Kaulfers, Chingyun Cheng, Xingyi Zhao, Balaji Selvaraj, Haibo Lin, Charles B. Simone, J. Isabelle Choi, Jenghwa Chang, Minglei Kang

**Affiliations:** 1Department of Physics and Astronomy, Hofstra University, 1000 Hempstead Turnpike, Hempstead, NY 11549, USA; glattery1@pride.hofstra.edu (G.L.); tkaulfers1@pride.hofstra.edu (T.K.); 2Department of Radiation Oncology, Rutgers Cancer Institute of New Jersey, 195 Little Albany Street, New Brunswick, NJ 08901, USA; chingyun.cheng@rutgers.edu; 3Beijing Key Laboratory of Medical Physics and Engineering, Peking University, Beijing 100871, China; daluo8@stu.pku.edu.cn; 4New York Proton Center, 225 E 126th Street, New York, NY 10035, USA; bselvaraj@nyproton.com (B.S.); hlin@nyproton.com (H.L.); csimone@nyproton.com (C.B.S.II); ichoi@nyproton.com (J.I.C.); 5Radiation Medicine, Donald and Barbara Zucker School of Medicine at Hofstra/Northwell, 450 Lakeville Road, Lake Success, NY 11042, USA

**Keywords:** FLASH radiotherapy, breast cancer, ultra-high dose rate, proton pencil beam scanning, single-energy Bragg peak, intensity-modulated proton therapy

## Abstract

**Simple Summary:**

Breast cancer is the most prevalent form of cancer worldwide and is projected to affect one in every eight American women over their lifetimes; radiation therapy (RT) is utilized for most breast cancer patients’ treatments. Our study aimed to evaluate a novel Bragg peak, ultra-high dose rate (FLASH), proton radiotherapy (PRT) technique that can potentially decrease radiotherapy toxicities. Clinically viable Bragg peak FLASH treatment plans were generated for breast cancer patients who were treated with conventional PRT. Across dosimetric standards of care for healthy and cancerous tissues, we observed feasible breast cancer treatment delivery and high plan quality while maintaining an ultra-high dose rate. Therefore, delivering proton FLASH-RT for breast cancer may reduce toxicities and improve the therapeutic ratio.

**Abstract:**

Bragg peak FLASH-RT can deliver highly conformal treatment and potentially offer improved normal tissue protection for radiotherapy patients. This study focused on developing ultra-high dose rate (≥40 Gy × RBE/s) intensity-modulated proton therapy (IMPT) for hypofractionated treatment of early-stage breast cancer. A novel tracking technique was developed to enable pencil beaming scanning (PBS) of single-energy protons to adapt the Bragg peak (BP) to the target distally. Standard-of-care PBS treatment plans of consecutively treated early-stage breast cancer patients using multiple energy layers were reoptimized using this technique, and dose metrics were compared between single-energy layer BP FLASH and conventional IMPT plans. FLASH dose rate coverage by volume (V_40Gy/s_) was also evaluated for the FLASH sparing effect. Distal tracking can precisely stop BP at the target distal edge. All plans (*n* = 10) achieved conformal IMPT-like dose distributions under clinical machine parameters. No statistically significant differences were observed in any dose metrics for heart, ipsilateral lung, most ipsilateral breast, and CTV metrics (*p* > 0.05 for all). Conventional plans yielded slightly superior target and skin dose uniformities with 4.5% and 12.9% lower dose maxes, respectively. FLASH-RT plans reached 46.7% and 61.9% average-dose rate FLASH coverage for tissues receiving more than 1 and 5 Gy plan dose total under the 250 minimum MU condition. Bragg peak FLASH-RT techniques achieved comparable plan quality to conventional IMPT while reaching adequate dose rate ratios, demonstrating the feasibility of early-stage breast cancer clinical applications.

## 1. Introduction

Radiation therapy (RT) is prominently utilized in the treatment of all stages of breast cancer [1] and has evolved significantly in recent years. For early-stage breast cancer cases, the standard radiation approach is treatment of the whole breast followed by a boost to the tumor bed or partial breast irradiation alone. Linear accelerator (LINAC)-based RT utilizing high-energy X-rays with or without electrons is the most common treatment option. When treating a superficial boost target, electrons have limited penetration depth, allowing for improved sparing of organs-at-risk (OARs) beyond the treatment target. When the tumor bed is at depth, conventional X-ray treatment, on the other hand, is a better option, even though the surrounding normal tissues can still receive a significant dose from the entrance and exit portions of the beam [2].

Proton radiotherapy (PRT) offers a further unique set of advantages and is clinically used as a treatment option for various types of cancer, including breast cancer. Compared to traditional radiation therapy techniques, proton therapy has the advantage of delivering radiation more precisely to the tumor, thereby minimizing damage to surrounding healthy tissues. PRT has shown promise, particularly in certain cases where there is a need to minimize radiation exposure to the heart and lungs, which are organs located close to the breast area [3,4,5,6,7,8]. This is particularly relevant for left-sided breast cancers, where the heart is located ipsilaterally (on the same side). By using proton therapy, the radiation dose to the heart and lungs can be reduced, lowering the risk of long-term complications such as radiation-induced heart disease and lung problems [9]. This may contribute to lower rates of long-term cardiac and pulmonary toxicities, which are especially important for younger patients and those with pre-existing heart or lung conditions. The use of PRT for partial-breast irradiation (PBI) and whole-breast irradiation (WBI) in the treatment of breast cancer is still an area of active research and clinical investigation. PBI aims to deliver radiation therapy specifically to the region of the breast where the tumor was removed, rather than irradiating the entire breast, which is typically delivered after breast-conserving surgery (BCS). Early investigations demonstrate excellent control rates and low rates of toxicities with proton therapy for early-stage breast cancer [10,11], and further research is needed to concretely establish the long-term outcomes and efficacy, optimal patient selection criteria, safety, and comparative benefits of PRT for PBI and WBI.

The inherent advantages of PRT are due to the proton beam’s finite range and its ability to deposit most of its energy at a precise depth known as the Bragg peak (BP). Beyond this peak, the energy deposition rapidly decreases, allowing for potentially far superior dose distribution and sparing healthy tissues beyond the target volume. With careful treatment planning, uncertainties in determining the exact range of the proton beam can be minimized for highly conformal treatment, accounting for factors such as tissue density variations, patient setup, and anatomical changes during treatment that could influence the range. The use of the BP techniques, including layering multi-energy peaks to create a spread-out Bragg peak (SOBP), allows the proton beam to effectively terminate within the patient during treatment. Compared to photon radiotherapy modalities, the complete removal of the exit dose makes PRT ideal when trying to avoid delivering doses past the target.

### 1.1. FLASH Background

PRT systems in clinical use allow for an innovative technique that delivers ultra-high dose rates of radiation in a fraction of a second, potentially reducing treatment times and further minimizing normal tissue toxicities [12]. Preliminary studies have shown promising results in preclinical models, but further research is needed to evaluate its safety and efficacy in clinical settings [13,14,15,16,17,18]. The novel ultra-high dose rate FLASH technique has been proposed as ≥40 Gy/s in its conception [13], compared to conventional proton radiation therapy, typically delivering ≤ 0.03 Gy/s dose rate. Most FLASH studies adopted this convention of a 40 Gy × RBE/s threshold; however, research has yet to find a comprehensive relation or lower dose limit to the threshold that can maintain the FLASH effect. Preliminary evidence shows ultra-high dose rates delivered below the set dose rate threshold for calculations can still meaningfully contribute to healthy tissue sparing [15]. The importance of the absolute dose delivered at measured dose rates is yet to be fully studied.

Cancer treatment with FLASH-RT has shown beneficial biological effects in vivo [19]. Preclinical studies have indicated that FLASH treatments can maintain similar tumor control while improving normal tissue sparing compared to conventional dose rate treatments. Several key factors [14] have been hypothesized as possible causes of the FLASH-effect, but biological mechanisms for the observed healthy tissue sparing are still under investigation. One major hypothesis is that the more rapid radiation delivery may not allow for healthy tissues to reoxygenate, resulting in increased radioresistance and better sparing of OARs [13]. Studies show fewer dicentric chromosome aberrations are created during FLASH compared to conventional dose rate treatments, suggesting differences in DNA damage response. Another possible contributing factor is the greater difference in cell cycle arrest during the G2 phase of treated cells that are actively proliferating [13].

### 1.2. Current Limitations

Initially, researchers mainly used modified LINACs to generate ultra-high dose rate electron beams and conduct biological FLASH experimental studies [20]. The delivery of FLASH with proton therapy is increasingly drawing attention due to its superiority in dose conformity compared to electron therapy, as well as the modality’s ability to treat larger and deeper target volumes. Clinical proton therapy facilities based on cyclotron systems can typically provide very high nozzle beam currents (>100 nA) compared to clinically used LINAC setups. Current PRT systems in clinical use can reach the FLASH dose rate threshold while considering relative biological effect (RBE) scaling (≥40 Gy × RBE/s) for preclinical and clinical studies [13]. The first clinical trial of proton FLASH-RT delivered to 10 patients was based on proton transmission beam (TB) PRT to evaluate the safety of proton pencil beam scanning (PBS) FLASH-RT [21,22,23]. In TB treatment, only the entrance portion of the percent depth curve is used to treat the target, and the entire BP region of the curve is placed outside of the patient’s body.

Current biological evidence shows that a lower isodose level may not trigger the FLASH effect [23,24,25,26]; therefore, conformal FLASH planning should still follow as low as reasonably achievable (ALARA) principles for normal tissues. It is also critical for FLASH treatment planning to maintain intensity-modulated proton therapy (IMPT)-like dosimetric quality compared to the current proton treatment criterion. Most current proton FLASH-RT studies focus on TB techniques [27,28]. Without beamline modifications, TB techniques provide the most straightforward and efficient FLASH delivery [28,29]. This PRT treatment technique introduces an exit dose in the normal tissue beyond the target, which is not an optimal method for delivery compared to conventional dose rate proton treatments using SOBP techniques and increases the irradiation doses received by critical normal tissues beyond the target volume. Conventional IMPT is the most advanced form of proton therapy and can optimally spare critical structures for cancer patients [20,29,30]. However, it historically has not been feasible to reach FLASH dose rates with IMPT due to two major reasons: (1) proton beams with lower energies cannot reach a sufficiently high beam current for FLASH delivery; (2) energy layer switching takes at least a few hundred milliseconds, which prolongs the beam-on time and significantly reduces the mean dose rate to be much lower than the required 40 Gy/s dose rate [30,31].

The BP phenomenon in proton radiotherapy offers several advantages over TB techniques. By exploiting the unique energy deposition characteristics of protons, the BP can allow for precise dose delivery to the tumor while minimizing radiation exposure to surrounding healthy tissues, which results in a lower normal tissue complication probability. The sharp dose fall-off beyond the BP also allows for tighter dose conformity, which would enable clinicians to better shape the radiation dose to conform to irregularly shaped tumors. These benefits make PRT with BP delivery particularly valuable in treating tumors located close to critical structures and in pediatric patients, where minimizing the radiation dose to healthy tissues is crucial.

We have made breakthrough progress on the proton BP FLASH-RT technique to deliver a highly conformal FLASH dose rate in hypofractionated proton radiotherapy using a single-energy proton beam [29,32]. This method has been demonstrated to achieve a superior inherent normal tissue-sparing dose relative to TB proton FLASH-RT plans with satisfactory FLASH dose rate coverage.

### 1.3. Treatment Rationale

As the standard of care in breast cancer treatment, post-operative radiotherapy typically used a conventional fractionation regimen (between 1.8 Gy and 2 Gy per fraction). However, for the most current clinical practice, an increase in the fractional dose and fewer fractionations (>2 Gy per fraction), taking advantage of the low alpha/beta ratio of breast cancer cells, has become a new standard of care for early-stage breast cancer [33,34]. The delivery of hypofractionated radiotherapy is often limited by the risk of late toxicities, especially soft tissue fibrosis. FLASH-RT has shown promising outcomes in sparing the normal tissue—with numerous preclinical publications demonstrating that the FLASH effect can reduce normal tissue fibrosis while maintaining a comparable cell-killing effect in cancerous tissues [17]. Many studies have shown FLASH-RT preserved functionality in various sites [13,14,15,16,17,18], including the lung, skin, brain, and abdomen. Therefore, there is great interest in translating proton FLASH-RT for PBI. After BCS for breast cancer, radiotherapy is shown to reduce the risk of recurrences and improve overall survival [35]. Although, late toxicities associated with conventional dose rate radiation therapy can impact breast cosmesis and quality of life. Studies have shown acceptable toxicity and excellent local control and survival with accelerated schedules in a FAST trial [36] comparison between 5-fraction schedules with traditional 25-fraction schedules. They found similar toxicity and confirmed these results with a 10-year follow-up. However, it may be more beneficial to further shorten the treatment time in order to improve patient convenience, reduce health disparities, and potentially further improve tumor control. A matched-case analysis comparing HF5 with hypofractionation in 15 fractions (HF15) and patients receiving lymph node irradiation and simultaneous integrated boost (SIB) showed fewer acute toxicities in the HF5 group [36]. A phase II trial, European REQUITE [37], investigated five-fraction once-weekly hypofractionated whole-breast irradiation (WH-WBI) as a treatment following BCS for stage 0-II breast cancer. The study aimed to assess the efficacy and safety of WH-WBI and compare disease-specific outcomes to conventional radiation techniques. Disease-specific outcomes after WH-WBI were found to be favorable and comparable to conventional radiation techniques for stage 0-II breast cancers. Previous studies have shown that WH-WBI could offer a more convenient and cost-effective alternative to conventional radiation techniques [37].

We hypothesize that BP FLASH-RT delivers doses with unique proton properties that can better protect normal tissues. The advancement of this method offers possibilities to optimize and accelerate the translational research of FLASH-RT. This study focuses on developing advanced FLASH-RT methods and preclinical protocols to accelerate the implementation of novel ultra-high dose rate BP techniques for PBI.

## 2. Materials and Methods

This study was reviewed and approved by the institutional review board. Ten consecutive left-sided partial-breast cancer patients who received traditional PBS IMPT post BCS were replanned for Bragg peak FLASH PRT. The clinical target volumes (CTVs) ranged from 72.7 cc to 338.4 cc, with a median of 134.3 cc. A hypofractionated 8 Gy × 5 prescription dose was adopted for a 40 Gy total dose delivered via FLASH treatment planning. Each conventional IMPT plan was developed considering both CTV coverage and dose to critical OARs, including ipsilateral left breast, ipsilateral left lung, heart, and skin. Similar considerations were applied when developing the FLASH-RT plans using the single-energy layer BP FLASH techniques. All plans generated were developed under clinical machine parameters using an in-house development treatment planning system (TPS) specifically created for FLASH-RT. The treatment plans were reviewed by clinical physicists and radiation oncologists to ensure alignment with clinical standards. The clinical goals for treatment planning remained consistent from patient to patient. However, the patient organ geometry and tumor sizes varied significantly, requiring individualized inverse treatment planning. CTV coverage and dose to OARs of BP FLASH plans were calculated and compared with those of conventional IMPT plans. A *p*-value less than 0.05 was defined as statistically significant for the analysis of the treatment plan data sets using a two-tailed paired t-teston the sample data for each metric.

### 2.1. Treatment Planning

Bragg peak FLASH plans were optimized using inverse treatment planning. The highest cyclotron energy of 250 MeV proton beam data allowed us to generate FLASH-RT treatment plans under a hypofractionated regimen (8 Gy × 5). With the in-house TPS, spot spacing was optimized to 5 to 8 mm to achieve dose uniformity and target coverage. All clinical PBS plan beam arrangements were adopted to optimize a direct comparison of plans, no couch angles were used before optimization, and all the final dose calculations applied a constant RBE of 1.1, scaling dose, and dose rate calculations.

For some plans, two additional regions of interest (ROIs), a 3 mm and 5 mm uniform expansion of CTV (CTV + 3 mm, CTV + 5 mm), were created to conform the spot distribution to the tumor geometries. The spot map optimization adopted in previous studies [29] was also used to improve the dose distribution and FLASH dose rate coverage, two competing terms both highly relying on the number of spots [29,32]. In general, a higher number of spots will improve the dosimetric distribution and decrease the FLASH dose rate coverage. The developed spot map optimization method tries to minimize the number of spots while maintaining the quality of the treatment plans to the greatest extent possible.

### 2.2. Dose Evaluation

Conventional PBS plans of these 10 patients developed using Eclipse TPS were compared to the FLASH treatment plans. All FLASH and conventional PBS plans were normalized so that at least 95% of CTV received 100% of the prescription dose, and dose-volume histograms (DVHs) were generated and exported to extract clinical dose metrics. Each plan was evaluated using 15 dose metrics for tumor coverage and healthy tissue-sparing. The dose metric data sets for all the 10 optimized plans were averaged for dosimetric analysis.

### 2.3. Dose Rate Evaluation

The equations used to calculate the minimum monitor unit (${MU}_{min}$) and beam current ($I_{beam}$) when computing dose and dose rates along with spot dwell time ($T_{spot}$) are given by [38]:(1)$${MU}_{min}=\frac{T_{MST}\times I_{beam}}{N_{proton}}$$
(2)$$T_{spot}=\frac{{MU}_{spot}}{{MU}_{min}}\times T_{MST}$$
where $T_{MST}$ is minimum PBS spot time, $N_{proton}$ is the number of protons beam delivered in 1 MU, and ${MU}_{spot}$ is the minimum monitor units delivered for any given spot. The average dose rate (ADR) was adapted for calculation by taking the difference in the total dose deposited at some a voxel J, $D_{J}$ over time $t_{1}-t_{o}$, where $t_{o}$ and $t_{1}$ are the start and stop times for voxel J in the scanning field [38]:(3)$${ADR}_{J}=\frac{D_{J}(t_{1})-D_{J}(t_{o})}{t_{1}-t_{o}}$$
With data generated for *D_J_*, the total dose deposited in some voxel J, and dose rate by voxel, ${ADR}_{J}$. The smaller the dose rate threshold, the stricter the dose constraint and dose deposited for ADR. All dose rate data were calculated with a small *d* of 0.1 Gy; more details can be found in reference [38]:(4)$$D_{J}\left(t_{1}\right)=D_{J}-d$$
(5)$$D_{J}\left(t_{o}\right)=d$$
where dose threshold *d* determines the effective cumulative dose needed to be delivered during PBS before the start time $t_{o}$ is taken. The clinical PBS speed used for treatment planning and calculation was 10 mm/ms. After incorporating the switch time, each spot can contribute to the ADR.

Some radiobiological experiments implied the FLASH sparing effect is more likely to be triggered when the single beam meets a minimal dose threshold [25,39]; therefore, dose thresholds of 0.1, 1, and 5 Gy were applied to assess the FLASH dose rate coverage for OARs. In a multiple-beam scheme for a hypofractionation regimen, two main factors contribute to the voxel-based dose rate statistics. Firstly, the beam switching time, including gantry rotation and other waiting time, is much longer than the delivery time. The time between beams should not be included in the dose rate calculation; thus, the dose rate of individual fields was calculated separately. Secondly, the dose threshold can either be applied for each field or for a plan. A dose rate volume histogram (DRVH) method was used to quantify the FLASH dose rate coverage for a plan, and only the voxels meeting a dose threshold (field dose or multiple field dose summed plan dose) were sampled to calculate DRVH [40]. FLASH coverage V_40Gy/s_ was derived from the percentage volume over a dose rate threshold of 40 Gy/s. More details of the dose rate coverage calculation method were reported in [20].

## 3. Results

### 3.1. Dosimetric Properties of Bragg Peak Plans

All BP FLASH plans were optimized following our clinical procedure and standards. A representative PBI case was replanned using Bragg-peak FLASH-RT techniques shown by field dose in Figure 1a–c, summed dose distribution in (e), and the conventional IMPT dose distribution in (f). The CTV volume is 140.1 cc (blue outlined ROI) in the breast (pink outlined ROI) shown in the transverse plane, showing thoracic cavity with critical heart (green outlined ROI) and ipsilateral left lung (light blue outlined ROI) OARs. An 8 Gy per fraction dose delivered to the CTV was achieved with individual fields with variable spot and beam weightings determined by inverse planning optimization. The DVH analysis, depicted in (d), demonstrated that both methods achieved similar target coverage. The single-energy Bragg peak plans exhibited a slightly higher dose concentration within the CTV and ipsilateral breast. On the other hand, the DVHs for OARs, such as the lung, heart, and skin, were almost identical for both techniques. Both plans utilized the Bragg peaks, IMPT with multiple energy layers, and FLASH with a single-energy layer.

Quantitative analyses and comparisons of PRT treatment plans for all 10 cases were then calculated, and plan quality was evaluated by comparing the dose metrics for tumor coverage and OARs. Figure 2 shows the results of the dose metric comparison of BP FLASH and IMPT treatment plans derived from DVH data. Each metric for BP and conventional IMPT plans was plotted and overlayed with each other using bar plotting. CTV V_100%_ represents the volume receiving 100% of the prescribed dose, and the BP was only about 1.3% higher than that of IMPT plans (*p*-value > 0.05), demonstrating that target conformity was similar between the two techniques. CTV V_105%_ in BP plans was about 32% higher than the IMPT ones. While the D_max_ of BP was slightly higher by ~4% compared to conventional IMPT, the D_max_ to the targets was within 115% of the prescription dose. The V_105%_ and D_2cc_ for the left breast were about 1% higher in BP plans, with a *p*-value less than 0.05. The D_max_ of skin for BP plans was about 13% higher than conventional IMPT plans, and the V_95%_ volume of the skin was about 7 cc more than that of IMPT plans. All these dose metrics showed a trend that the conventional IMPT plan can yield better dose uniformity or less high dose hot spots. The rest of the dosimetric values could not be distinguished between treatment plan types, including all heart and ipsilateral left lung dose metrics. See Appendix A for a numerical breakdown of each *p*-value and dose metric value average.

### 3.2. Dose Rate Coverage

Further plan quality assessment was performed for ultra-high dose rates. Figure 3 shows the 2D dose rate distribution of individual fields (a–c) from the same representative PBI case as well as the DRVH (d) and 2D dose rate distribution for a plan (e). The 2D dose rate calculation considered the delivery and spot time, which tends to have a higher dose rate at the field edge and a lower dose rate in the central area. When only considering voxels that received more than 5 Gy, there was no dose touching the heart, and only a small portion of lung volume close to the target received doses more than 5 Gy, as displayed in Figure 3f, demonstrating that even without considering any FLASH protection effect, the lack of an exit dose achievable with the BP technique can minimize the doses to heart and lung tissues. Accordingly, the V_40Gy/s_ coverage was increased for all the OARs, as shown in the DRVH in Figure 3d, when the 5 Gy threshold was applied compared to 0.1 Gy. The smaller volume of left lung tissue that received the dose also showed ultra-high dose rates (>40 Gy × RBE/s).

The dose rates of each OAR were calculated 3 × 2 times, including three dose thresholds of 0.1, 1, and 5 Gy and two different scenarios considering the dose thresholds for an individual field and plan dose, i.e., a voxel received dose from a single field should have been greater than the specific dose level or a voxel-received plan dose should have been greater than the specific dose threshold. All the dose rate statistics of the 10 cases were averaged to assess the dose FLASH coverage quantitively. Applying the dose thresholds for individual fields is a more conservative way to estimate the ultra-high dose rate coverage, as the FLASH coverage is evaluated field by field. As shown in Figure 4, the averaged V_40Gy/s_ coverage changed as a function of dose thresholds, and when higher dose thresholds were applied, higher FLASH ratios were demonstrated. When the 5 Gy dose threshold was applied, the V_40Gy/s_ ratio was increased to greater than 60% for the ipsilateral lung, breast, and skin for single-field evaluation. Only very low doses reached the heart, and there was no dose higher than 5 Gy for the heart. Similarly, the FLASH ratio increased from ~15% to ~35% when 0.1 and 1 Gy thresholds were applied for the heart.

## 4. Discussion

IMPT is one of the most advanced modalities for radiotherapy and has found extensive application in treating a wide array of tumors suitable for radiation therapy. IMPT employs the PBS delivery technique and administers doses to the tumor through a layer-by-layer energy delivery approach. In comparison to photon or scattering proton therapy, IMPT exhibits superior dosimetric attributes. However, ultra-high dose rate FLASH has been impossible for standard-of-care PBS treatment using SOBP due to the inability to switch between multiple energy layers within the delivery timeframe; however, the single-energy Bragg peak delivery can still reach the FLASH dose rate.

The novel proton Bragg peak FLASH technique demonstrated acceptable dose distributions and constraints when compared to that of conventional PBS methods. The monoenergetic, 250 MeV beam was able to achieve sufficient FLASH dose rate coverage for all patients, allowing for the expected tissue-sparing effects to exist. Inverse treatment planning, with spot map optimization, separate dose, and dose rate analysis software, produced data sets to come to conclusions on plan quality. Results with equivalent plan qualities to conventional plans were achieved and demonstrated through clinically approved dose metrics as well as dose rate coverage through the OARs.

FLASH ultra-high dose rate use in PRT has been observed to offer unique biological and healthy tissue-sparing effects. Current FLASH studies use TB; however, developments using distal tracking allow the use of BP FLASH that can be precisely stopped at the target distal edge. The combination of these techniques is shown to result in tumor coverage and dose conformality equivalent to IMPT plans coupled with the potential biological advantages of FLASH-RT plans. Inverse treatment plans generated by simulating FLASH techniques for partial breast cancer treatment in a clinical setting were largely comparable to conventional PBS proton RT (*n* = 10). Most dose metrics for the CTV and OARs evidenced an inability to distinguish the two data sets between plan types using a statistical *t*-test (*p* > 0.05). Notably, the heart and ipsilateral left lung dose distributions were IMPT-like in all FLASH-RT treatment plans. Dose metrics where FLASH-RT treatment planning slightly outperformed conventional PBS treatment were statistically insignificant; however, most dose metrics where IMPT plans outperformed FLASH were also statistically insignificant.

For high doses delivered to small volumes, clinically refined IMPT plans did show superior dose uniformity compared to FLASH plans in superficial left breast and skin OARs, which is consistent with the previous observation in the lung [20], liver [31], and head–neck cases [41]. This is because the conventional IMPT plans use multiple energy layers. In contrast, BP plans only use single energy to achieve dose uniformity, lacking flexibility in energy tuning, making dose uniformity slightly inferior. Considering the ultra-high dose rate in BP plans, the FLASH-sparing effect might offer improved protection in skin tissue, making the high doses less concerning [42,43]. Even though the inherent sparing benefits of FLASH have the potential to mitigate dose variation, the elevated doses to the skin and CTV require further refinement and warrant additional studies in treatment planning prior to clinical implementation.

Table A1 shows that dose metrics required for quality treatment delivery were achieved. Most dosimetrics found no statistically discernable differences between data sets using a paired t-test. A lack of statistical difference between IMPT and FLASH in the heart dose and left lung indicates that Bragg-peak FLASH at least will not increase the complication probability of these two organs. Additionally, the likelihood of complications may even decrease, depending on the extent to which the FLASH dose and dose rate can be attained for these two organs. Considering that the biological model for the FLASH effect is still in development, a qualitative reduction could be anticipated.

Analyzing the threshold for FLASH coverage, various dose rate calculations for the ≥40 Gy × RBE/s threshold were assessed to determine if FLASH-effects could be achieved and for what ROIs. With a set dose rate threshold, dose thresholds were also implemented to determine if FLASH sparing was in effect for the most important dose contributions. An in vitro study demonstrated such thresholds, with the effects of FLASH seen beginning at 5 Gy and becoming clear at 15 Gy with significance at 18 Gy [39]. The biological relation for these dosages is how much oxygen depletion is seen in OARs at these dose levels. The increase in severity of the cellular hypoxia increases radioresistance in cells and can allow for more inherent protection and sparing of the OARs. This may create some dilemmas for OARs that have lower dose constraints but would ideally receive more definitive FLASH effects by crossing certain thresholds. The balance of these effects is left for further studies. Table A2 evidences that sufficient dose rates were achieved for the FLASH effect of sparing healthy tissue. The statistics for V_40Gy/s_ FLASH coverage across all 10 cases exclude voxels by implementing minimum dose thresholds of 0.1, 1, and 5 Gy. These mean values and their corresponding standard deviations are calculated from the average V_40Gy/s_ of all ten patients.

When discussing inverse treatment planning and its parameters, there are many factors that can heavily determine plan quality and success, but one that is crucial for this study is the minimum spot time $\left(T_{MST}\right)$. We used a practical machine time $T_{MST}$ of 0.5 ms. Minimum MU $({MU}_{min}$) is also an important factor when it comes to dose delivery. In fact, using a larger min MU can improve the FLASH ratios. For PBI treatment plans generated under FLASH dose rates, ${MU}_{min}$ was kept constant at a low value of 250 MU, but it could be increased in future FLASH-RT treatment planning for improved FLASH dose rate coverage if proven necessary. Other variables such as beam angles and dose constraints were altered patient by patient to generate plans as similar to conventional plans as possible.

## 5. Conclusions

The proposed single-energy Bragg peak FLASH technique eliminates the exit dose associated with transmission proton FLASH, while still achieving comparable plan quality and OAR sparing, and while maintaining sufficient FLASH dose rate coverage for breast proton hypofractionated radiotherapy. This study demonstrated the feasibility of the single-energy Bragg peak FLASH-RT treatment for early-stage partial breast cancer treatments, which could lead to clinical implementation.

## Figures and Tables

**Figure 1 cancers-15-04560-f001:**
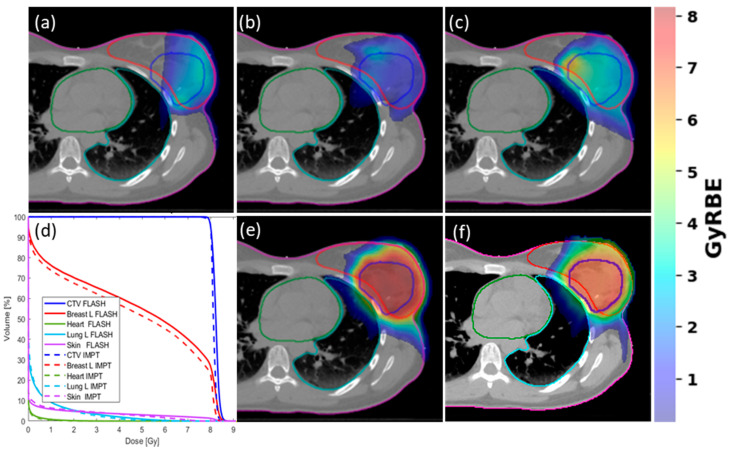
Application of single-energy Bragg peak optimization can achieve ultra-high FLASH dose rate and IMPT-like plan quality: 2D dose distribution from FLASH single beam at gantry angle of (**a**) 0 degrees, (**b**) 70 degrees, and (**c**) 125 degrees. (**d**) Normalized DVH (95% of CTV receiving prescription dose) plot comparison of the FLASH (solid lines) and IMPT (dashed lines) treatment plans; (**e**) summed 2D dose distribution for the FLASH plan; (**f**) summed 2D dose distribution for the corresponding IMPT plan.

**Figure 2 cancers-15-04560-f002:**
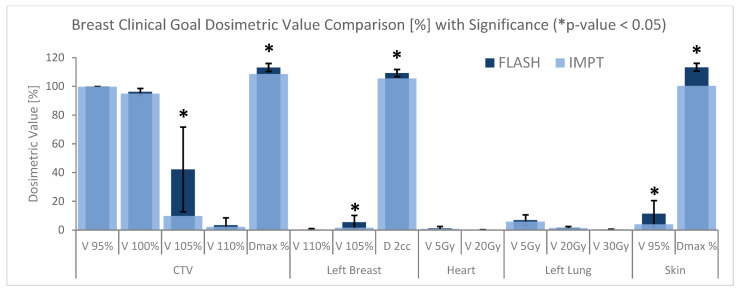
Overlapping histogram of absolute dosimetric value comparisons between FLASH and IMPT treatment plans. Dose metric values bars without asterisk notation found plan sample data sets statistically insignificant from each other. The error bars represent the standard deviation for FLASH dosimetrics; complete standard deviation values can be found in Table A1.

**Figure 3 cancers-15-04560-f003:**
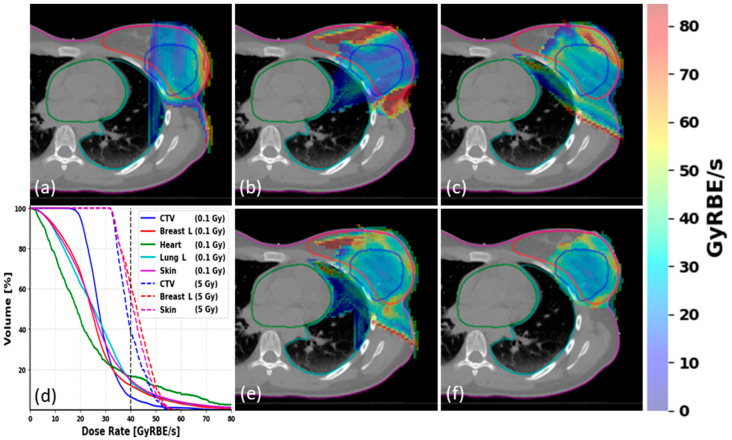
Bragg peak FLASH-RT treatment plans dose rate characteristics: 2D dose rate distribution from FLASH single beam at gantry angle of (**a**) 0 degrees, (**b**) 70 degrees, and (**c**) 125 degrees. (**d**) Merged field DRVH plot for plan dose thresholds of 0.1 and 5 Gy, and the vertical dash line represents 40 Gy × RBE/s dose rate; (**e**) 2D dose rate distribution when applying plan dose threshold of 0.1 Gy; (**f**) 2D dose rate distribution excluding voxels receiving less than 5 Gy total dose.

**Figure 4 cancers-15-04560-f004:**
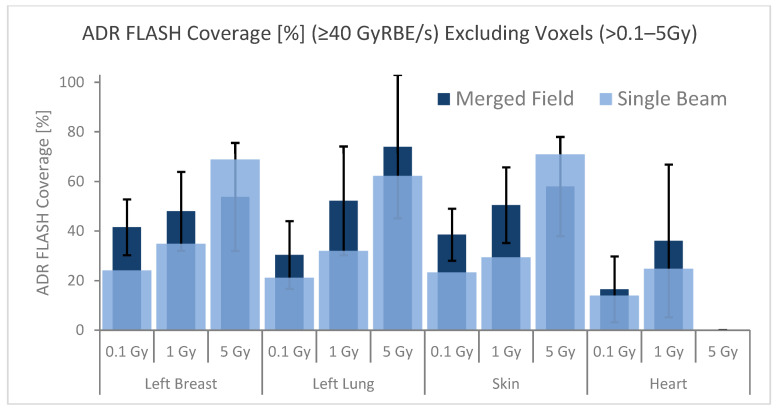
V_40 Gy/s_ FLASH percentage of OARs. Dose rates of each OAR were calculated 3 × 2 times, with 3 dose threshold levels of 0.1, 1, and 5 Gy under 2 different scenarios, i.e., dose thresholds were considered for plan and individual field dose. OARs receiving less dose than the dose exclusion show no data, e.g., greater than 5 Gy for the heart. The error bar represents the dose rate standard deviation from the mean values for plan (summed fields); complete standard deviation values can be found in Table A2.

## Data Availability

The research data will be shared upon request to the corresponding authors.

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
