# Peer review of "Pencil Beam Scanning Bragg Peak FLASH Technique for Ultra-High Dose Rate Intensity-Modulated Proton Therapy in Early-Stage Breast Cancer Treatment"

_cancers, 2023, doi:10.3390/cancers15184560_

Round 1

Reviewer 1 Report

Very interesting and novel topic..

Please revise references according to Cancer style.

Good quality of English, minor changes required.

Author Response

Comments and Suggestions for Authors

Very interesting and novel topic.

Authors: Thank you. We are grateful for the time the reviewers dedicated to assessing our manuscript.

Please revise references according to Cancer style.

Authors: Thank you for the reminder. References made self-consistent,  and we will adjust following specific format following journal guidelines.

Comments on the Quality of English Language

Good quality of English, minor changes required.

Authors: A broad editing pass was completed. Thank you for your contribution.

Reviewer 2 Report

  The article is overall well written, methods are sound and correctly described and the topic (the potential clinical adoption of FLASH radiotherapy) is up to date and of great interest. Nonetheless, I have a few minor remarks.

- as described in the results, the main drawback of the proposed technique compared with IMPT is the worse homogeneity of target dose. Although non significant, the increase of V105 Is relevant and in my clinical practice I would actually not accept such percentages of V105. Higher skin dose, even if less remarkably, is obtained as well. Therefore, I would better explain in the discussion section how this pitfalls are counterbalaced by the advantages of FLASH RT and what this advantages are. This would support the claim of and increased therapeutic ratio

- material and methods: explain why you adopted a 8 Gy x 5 schedule which, to the best of my knowledge, is not among the most commonly used APBI schedules. Specify as well  what the actually erogated schedule was. Moreover I would specify the adopted dose constraints.

Minor corrections
- Page 2 line 48: replace one of the two "treatment"
- FLASH background: the potential biologic advantage of FLASH should be briefly explained
- Page 4 line 158: as correctly stated subsequently, the current standard of care Is actually hypofractionated RT
- Page 4 line 176: rephrase "more optimal"
- Page 5 line 230: has PTV dose been evaluated as well?
- Page 6 line 253: complete the sentence "more details can be found..."
-results: maybe I would move the example case after the main results
-page 9 line 346: replace > sign with complete wording
- line 367: rephrase "Without
having to spend time switching"             

Minor editing of English language required

Author Response

We appreciate the time that the reviewers dedicated to evaluating our manuscript.

Comments and Suggestions for Authors

The article is overall well written, methods are sound and correctly described and the topic (the potential clinical adoption of FLASH radiotherapy) is up to date and of great interest. Nonetheless, I have a few minor remarks.

Authors: Thank you. We are grateful for the time the reviewers dedicated to assessing our manuscript.

As described in the results, the main drawback of the proposed technique compared with IMPT is the worse homogeneity of target dose. Although non significant, the increase of V105 Is relevant and in my clinical practice I would actually not accept such percentages of V105. Higher skin dose, even if less remarkably, is obtained as well. Therefore, I would better explain in the discussion section how these pitfalls are counterbalaced by the advantages of FLASH RT and what this advantages are. This would support the claim of and increased therapeutic ratio

Authors: Thank you for the valuable comments and suggestions. For high doses delivered to small volumes, clinically refined IMPT plans did show superior dose uniformity compared to FLASH plans in superficial left breast and skin OARs. This is because the conventional IMPT plans use multiple energy layers. In contrast, BP plans only use single energy to achieve dose uniformity, lacking flexibility in energy tuning, making dose uniformity slightly inferior. Considering the ultra-high dose rate in BP plans, the FLASH-sparing effect might offer improved protection in skin tissue, making the hot doses less concerning. While the inherent sparing benefits of FLASH have the potential to mitigate dose variation, the elevated doses to the skin and CTV require further refinement and warrant additional studies in treatment planning prior to clinical implementation. This is now added to the Discussion from lines 404-407.

Material and methods: explain why you adopted a 8 Gy x 5 schedule which, to the best of my knowledge, is not among the most commonly used APBI schedules. Specify as well  what the actually erogated schedule was. Moreover, I would specify the adopted dose constraints.

Authors: Thank you for bringing it up. We agreed that this is not a typical clinical regimen for partial breast radiation.   At the moment, the biological mechanisms behind the FLASH effect are not well determined. Most of the preclinical biological studies used a single and a single field to deliver the whole prescribed dose. Therefore, all planning studies rely on certain assumptions that have not been fully validated based on evidence yet. Fortunately, there is more and more experimental evidence as to the growing interest in this topic. As fractionated radiotherapy regimens are the standard of care for the majority radiation treatments, Montay-Gruel et al. recently studied the hypo-fractionation FLASH efficacy for mice glioblastoma in mice. Different regimens 10 Gy, 2 x 7 Gy, and 3 x10 Gy all show the FLASH normal tissue sparing effect [1]. Their findings implicate, even not report yet, that FLASH-RT is delivered by multiple fields, the FLASH sparing effect will still exist. Therefore, the transmission planning studied by different research groups for head & neck and lung all used multiple-field to deliver uniform dose targets. Based on similar logic, we developed multiple-field optimized intensity-modulated plans using single-energy Bragg peaks to deliver conformal FLASH-RT to partial breast cancers. Our studies aims to conduct a thorough comparison between FLASH and the current PBS treatment techniques, showcasing the viability of treating partial breast patients using identical metrics. We acknowledge that determining the suitable regimen and dose metrics requires comprehensive investigations and clinical trial studies.

Reference:

[1] Montay-Gruel P, Acharya MM, Gonçalves Jorge P, Petit B, Petridis IG, Fuchs P, Leavitt R, Petersson K, Gondré M, Ollivier J, Moeckli R, Bochud F, Bailat C, Bourhis J, Germond JF, Limoli CL, Vozenin MC. Hypofractionated FLASH-RT as an Effective Treatment against Glioblastoma that Reduces Neurocognitive Side Effects in Mice. Clin Cancer Res. 2021 Feb 1;27(3):775-784. doi: 10.1158/1078-0432.CCR-20-0894. Epub 2020 Oct 15. PMID: 33060122; PMCID: PMC7854480.

Minor corrections
- Page 2 line 48: replace one of the two "treatment"

Authors: Thank you, deleted the first "treatment."

- FLASH background: the potential biologic advantage of FLASH should be briefly explained

Authors: Thank you. The can be found in lines 89-92 and 102-104.

- Page 4 line 158: as correctly stated subsequently, the current standard of care is actually hypofractionated RT

Authors: Thank you, rephrased using past tense for clarity.

- Page 4 line 176: rephrase "more optimal"

Authors: Thank you; it was rephrased.

- Page 5 line 230: has PTV dose been evaluated as well?

Authors: We only use CTV for dose optimization and evaluation for proton therapy at our institute.

- Page 6 line 253: complete the sentence "more details can be found..."

Authors: Thank you; it was corrected.

-results: maybe I would move the example case after the main results

Authors: Thank you for your suggestion. We attempted to rearrange the figures and discovered that presenting the example case before the average results resulted in a clearer narrative.

-page 9 line 346: replace > sign with complete wording

Authors: Thank you, complete wording added.

- line 367: rephrase "Without having to spend time switching"

Authors: The sentence was rewritten for clarity. Thank you for your contributions.

Reviewer 3 Report

The paper is explaining a relatively new concept in FLASH proton radiotherapy performed on 10 patients. The paper sounds scientific but has some significant flaws that has to be addressed before making further decisions.

besides that, I strongly believe that other journals dedicated to radiotherapy are more suitable for publishing this paper. 

below are some comments to improve the paper:

1. introduction is too lung and seems like a review paper. I suggest authors to concentrate all information in one-part introduction without sub titles.

2. the quality of figure 2 and 4 is low. also the figures need Y axes title

3.  what is the angle of gantry for DRVH plot in figure 3

4. despite introduction, the discussion is too short. 

5. delete Stereotactic body radiotherapy from keywords list

6. why you selected only 10 patients? explain the statistical method in the method and material! I think you need more patients for drawing a conclusion

7.  What was the treatment planning software (TPS)? what do you mean by n in-house development treatment planning software (TPS)

8. please explain who performed the treatment planning? did one person do all treatment planning? 

9. since we know that IMRT has higher efficiency in conventional radiotherapy, explain the novelty of your study?

10. based on the table A1, there is no difference between IMPT and FLASH in heart dose and left lung; please explain the superiority of IMPT over FLASH in this case then

11. Tables in Appendix need to be analysed in the discussion

12. explain the limitations of IMPT in the discussion

Author Response

Thank you to the reviewer for investing valuable time in evaluating our work. The comments and suggestions provided have enhanced the quality of our study.

Comments and Suggestions for Authors

The paper is explaining a relatively new concept in FLASH proton radiotherapy performed on 10 patients. The paper sounds scientific but has some significant flaws that has to be addressed before making further decisions.

Authors: Thank you. We are grateful for the time the reviewers dedicated to assessing our manuscript.

besides that, I strongly believe that other journals dedicated to radiotherapy are more suitable for publishing this paper. 

Authors: We have submitted this study to the special issue of Cancers dedicated to Particle Therapy, a category focused on radiation therapy. We are confident that this study aligns effectively with the theme of this special issue.

below are some comments to improve the paper:

  1. introduction is too lung and seems like a review paper. I suggest authors to concentrate all information in one-part introduction without sub titles.

Authors: Thank you for the suggestion. FLASH radiation therapy stands as a novel treatment modality with the potential to revolutionize the practice of Radiation Oncology. It is of utmost importance to distinctly outline the background and rationale behind the development of Bragg peak FLASH techniques for partial breast irradiation. We are of the opinion that a comprehensive review of the background in the introduction would be both beneficial and informative, particularly for audiences less acquainted with the subject.

  1. the quality of figure 2 and 4 is low. also the figures need Y axes title

Authors: Thank you for pointing it out, The two figures were regenerated with high quality.

  1. what is the angle of gantry for DRVH plot in figure 3

Authors: DRVH was generated from gantry angles 0, 70, and 125. This is now clarified in the figure legend.

  1. despite introduction, the discussion is too short. 

Authors: Thank you. Discussion was extended accordingly, and please find new contents highlighted in yellow color.

  1. delete Stereotactic body radiotherapy from keywords list

Authors: Thank you, keyword deleted.

  1. why you selected only 10 patients? explain the statistical method in the method and material! I think you need more patients for drawing a conclusion

Authors: An analysis of the treatment plan data sets was conducted using a two-tailed paired T-test on the sample data for each metric. We acknowledge that an increased number of patients would undoubtedly enhance statistical stability. Given the distinction in techniques between single-energy Bragg peak and multiple-energy PBS, a sample size of ten patients was considered substantial for this initial study to establish a robust conclusion.

  1. What was the treatment planning software (TPS)? what do you mean by n in-house development treatment planning software (TPS)

Authors:  Apologies for the confusion. The treatment planning software (TPS) is a tool to design radiation plans to deliver the prescribed dose by Radiation Oncologies. We have both commercial TPS Eclipse and an in-house developed tool for advanced research.

  1. please explain who performed the treatment planning? did one person do all treatment planning? 

Authors: Clarified that multiple researchers performed treatment planning.

  1. since we know that IMRT has higher efficiency in conventional radiotherapy, explain the novelty of your study?

Authors: IMRT is the most widely used technique in photon-based radiation therapy. An emerging approach is proton therapy. The innovation in our study lies in utilizing the single-energy Bragg peak of proton beams in conjunction with the ultra-high dose-rate FLASH criteria being fulfilled. This combination is a novel approach that has not been attempted previously and has demonstrated significant enhancement for FLASH proton therapy. This novelty is described in the following sentence of the “Simple Summary”:

“Our study aims to evaluate a novel Bragg peak, ultra-high dose-rate FLASH proton radiotherapy technique that can potentially decrease radiotherapy toxicities.”

  1. based on the table A1, there is no difference between IMPT and FLASH in heart dose and left lung; please explain the superiority of IMPT over FLASH in this case then

Authors: The absence of a statistical difference permits us to infer that FLASH can at least attain an equivalent dose distribution, particularly concerning the heart and left lung. The potential of the FLASH technique to induce more favorable biological effects holds promise for diminishing the toxicity and side effects on the heart and lungs caused by radiation. The following paragraph in lines ???-??? of the revised paper is added to emphasize this point:

“Appendix Table A1 shows that dose metrics required for quality treatment delivery were achieved. Most dosimetrics found no statistically discernable differences between datasets using a paired T-test. A lack of statistical difference between IMPT and FLASH in heart dose and left lung indicates that Bragg-peak FLASH at least will not increase the complication probability of these two organs. Additionally, the likelihood of complications may even decrease, depending on the extent to which the FLASH dose and dose rate can be attained for these two organs. Considering that the biological model for the FLASH effect is still in development, a qualitative reduction could be anticipated.”

  1. Tables in the Appendix need to be analyzed in the discussion

Authors: Thank you for pointing this out, and the analysis was added now.

  1. explain the limitations of IMPT in the discussion

Authors: The following paragraph is added at the beginning of the “Discussion” to address this comment:

“IMPT currently stands as the standard of care and has found extensive application in treating a wide array of tumors suitable for radiation therapy. IMPT employs the PBS delivery technique and administers doses to the tumor through a layer-by-layer energy delivery approach. In comparison to photon or scattering proton therapy, IMPT exhibits superior dosimetric attributes. However, ultra-high dose-rate FLASH has been impossible for standard-of-care PBS treatment using SOBP due to the inability to switch between multiple energy layers within the delivery timeframe; however, the single-energy Bragg peak delivery can still reach FLASH dose rate.”

Round 2

Reviewer 3 Report

The paper has been improved significantly but still :

1. figures don't have high quality for publication specially figures 2 and 4. remove lines from figures 2 and 4

please take into account that you want to publish in Q1 journal

2. the treatment planning has been done by several people. how are you aware of consistency in treatment plans then? what about pre tissue contouring? 

Author Response

The lines in Figures 2 and 4 have been removed. We double-checked the image quality in Word format, which appears to be good. I assume the reviewer requested the images in PDF format, but this may compromise the quality when exported from Word to PDF. Additionally, we have corrected the inconsistent colors in Figure 1, as pointed out by the editor. Furthermore, we have addressed the question regarding the quality of the treatment planning and attached the necessary clarification.
-------------------------------------------------------------------------------------------

The treatment planning has been done by several people. how are you aware of consistency in treatment plans then? what about pre tissue contouring?

Authors: We have one researcher dedicated to planning, with two certified medical physicists highly involved in some of the planning as well as in all the plan design and case-by-case evaluation.  All the patients were clinical cases; thus, the MD approved the contours for targets and OARs before this study.  To avoid any confusion, we have added new content to mention how to control the quality of planning: The treatment plans were reviewed by clinical physicists and radiation oncologists to ensure alignment with clinical standards.

As the reviewer and editor are aware, in clinical practice, the quality and uniformity of treatment plans can vary between different centers and individual planners. However, there are always clinical standards or protocols in place to control quality and uniformity. Our practice includes a team of 11 dosimetrists, all of whom have received thorough training in treatment planning for breast cases. Additionally, we have several radiation oncologists who specialize in treating breast cancers. It is rare for a single dosimetrist to work on breast planning in isolation or to collaborate with only one physician. We maintain the quality of treatment plans by applying clinical metrics.

Similar to the Bragg peak research plans, we involve clinical physicists in the planning design and optimization, and both clinical physicists and physicians participate in dosimetric assessment. The goal of Bragg peak planning is to closely match the dose metrics of conventional IMPT. We have confidence in the quality of the Bragg peak planning for two reasons. Firstly, the Bragg peak plan follows a similar beam arrangement to conventional PBS plans. Secondly, the plan quality of the two techniques is comparable when comparing dose metrics.

We believe that there is no single "best practice," but we consistently aim for optimal plan quality that aligns with the metrics implemented in the clinic. Treatment planning always has the potential for optimization, depending on the techniques, software, and individuals' experience. Clinical judgment can sometimes be subjective, and this study serves as a proof of concept for the novel technique in breast applications. We anticipate that a commercial TPS will become available, enabling more researchers to explore the dosimetric characteristics of the Bragg peak FLASH.

In Figure 1, the colors of structures in DVHs and dose maps are not consistent across subplots. Please revise.  

Authors: It has been corrected. Thank you for pointing this out.